# Partner-Inflicted Brain Injury: Intentional, Concurrent, and Repeated Traumatic and Hypoxic Neurologic Insults

**DOI:** 10.3390/brainsci15050524

**Published:** 2025-05-19

**Authors:** Julianna M. Nemeth, Clarice Decker, Rachel Ramirez, Luke Montgomery, Alice Hinton, Sharefa Duhaney, Raya Smith, Allison Glasser, Abigail (Abby) Bowman, Emily Kulow, Amy Wermert

**Affiliations:** 1College of Public Health, The Ohio State University, Columbus, OH 43210, USA; clarice.m.edw@gmail.com (C.D.); lukemonty7@gmail.com (L.M.); duhaney.3@buckeyemail.osu.edu (S.D.); smith.12969@osu.edu (R.S.); amg632@ints.rutgers.edu (A.G.); abbybowman2@gmail.com (A.B.); wermert.7@osu.edu (A.W.); 2Ohio Domestic Violence Network, Columbus, OH 43215, USA; rachelr@odvn.org (R.R.); emilyk@odvn.org (E.K.)

**Keywords:** domestic violence (DV), strangulation, hypoxic–anoxic brain injury (HAI), head trauma, traumatic brain injury (TBI), intimate partner violence (IPV), partner-inflicted brain injury (PIBI), brain injury from violence (BI-V), trauma-informed care (TIC)

## Abstract

(1) Background: Traumatic brain injury (TBI) is caused from rapid head acceleration/deceleration, focal blows, blasts, penetrating forces, and/or shearing forces, whereas hypoxic–anoxic injury (HAI) is caused through oxygen deprivation events, including strangulation. Most service-seeking domestic violence (DV) survivors have prior mechanistic exposures that can lead to both injuries. At the time of our study, some evidence existed about the exposure to both injuries over the course of a survivor’s lifetime from abuse sources, yet little was known about their co-occurrence to the same survivor within the same episode of physical intimate partner violence (IPV). To better understand the lived experience of service-seeking DV survivors and the context in which partner-inflicted brain injury (PIBI) is sustained, we sought to understand intentional brain injury (BI) exposures that may need to be addressed and accommodated in services. Our aims were to 1. characterize the lifetime co-occurrence of strangulation and intentional head trauma exposures from all abuse sources to the same survivor and within select physical episodes of IPV and 2. establish the lifetime prevalence of PIBI. (2) Methods: Survivors seeking DV services in the state of Ohio in the United States of America (U.S.) completed interview-administered surveys in 2019 (*n* = 47). Community-based participatory action approaches guided all aspects of the study development, implementation, and interpretation. (3) Results: The sample was primarily women. Over 40% reported having Medicaid, the government-provided health insurance for the poor. Half had less than a postsecondary education. Over 80% of participants presented to DV services with both intentional head trauma and strangulation exposures across their lifetime from intimate partners and other abuse sources (i.e., child abuse, family violence, peer violence, sexual assault, etc.), though not always experienced at the same time. Nearly 50% reported an experience of concurrent head trauma and strangulation in either the first or last physical IPV episode. Following a partner’s attack, just over 60% reported ever having blacked out or lost consciousness—44% experienced a loss of consciousness (LOC) more than once—indicating a conservative estimate of a probable brain injury by an intimate partner. Over 80% of service-seeking DV survivors reported either a LOC or two or more alterations in consciousness (AICs) following an IPV attack and were classified as ever having a partner-inflicted brain injury. (4) Conclusions: Most service-seeking IPV survivors experience repetitive and concurrent exposures to abusive strangulation and head trauma through the life course and by intimate partners within the same violent event resulting in brain injury. We propose the use of the term partner-inflicted brain injury (PIBI) to describe the physiological disruption of normal brain functions caused by intentional, often concurrent and repeated, traumatic and hypoxic neurologic insults by an intimate partner within the context of ongoing psychological trauma, coercive control, and often past abuse exposures that could also result in chronic brain injury. We discuss CARE (Connect, Acknowledge, Respond, Evaluate), a brain-injury-aware enhancement to service delivery. CARE improved trauma-informed practices at organizations serving DV survivors because staff felt knowledgeable to address and accommodate brain injuries. Survivor behavior was then interpreted by staff as a “can’t” not a “won’t”, and social and functional supports were offered.

## 1. Introduction

### 1.1. Prevalence and Health Impact of Intimate Partner Violence (IPV)—Worldwide

Around the world, violence has been used by groups with institutionalized power to terrorize and reify political, economic, and social domination [1,2,3]. The World Health Organization (WHO) defines intimate partner violence (IPV) as “behavior within an intimate relationship…by both current and former spouses and partners…that causes physical, sexual or psychological harm, including acts of physical aggression, sexual coercion, psychological abuse and controlling behaviors” [4]. The IPV risk is higher within relationships where one or both partners represent groups denied institutional power.

For example, the WHO estimates, from data extracted to the WHO Global Database on Prevalence of Violence Against Women, that worldwide, in their lifetime, 27% of ever married/partnered women aged 15 years and older have been targeted for physical and/or sexual IPV [4]. However, there is a wide variation in the IPV prevalence dependent on country or region—with a range of 16% in southern Europe to 40% or higher in Central Africa and parts of Oceania [5]. The odds of IPV victimization are 5.8 times (×) higher among individuals seeking refugee or political asylum, 4.2× higher among those not following traditional gender role expectations, and 3.2× higher among ethnic minorities [6]. The odds of IPV victimization are 2.5× higher in communities where there is a high rate of community violence and experiences of community violence, 2.8× higher when community norms are supportive of violence, and 2.1× higher in societies exposed to war or political violence.

IPV is associated with a wide range of disparate health outcomes [4]. For example, in comparison to those not subjected to abuse, IPV survivors worldwide have higher odds of poor general health (1.9×); mental and neurological disorders (2.7×), including impairments in offspring neurodevelopment; impaired cognitive and academic performance (3.1×); and poor daily function (2.9×), including in relation to tasks of daily living and mobility, an indication of disability.

### 1.2. The Prevalence and Health Impact of IPV—A United States (U.S.) Perspective

In the United States (U.S.), where there are substantial wealth and status divides, the lifetime exposure to IPV is also more prevalent among populations oppressed through historical and ongoing institutionalized domination [7]. According to The National Intimate Partner and Sexual Violence Survey, a representative sample of the non-institutionalized U.S. population, 42% of women will be targeted for physical IPV and almost 20% for contact sexual IPV. Though the physical IPV exposure is comparable for men, contact intimate partner sexual violence is less than half of that for women [8]. Women experience the disproportional burden of detrimental outcomes. For instance, 41.0% of women versus 26.3% of men report IPV with lasting social-, financial-, safety-, and health-related impacts.

In the U.S. context, in addition to the factors mentioned above, race has also shaped violence exposure, including IPV [7]. At the founding of the U.S., the construct of race was created as a visual way to justify dehumanization based on skin tone and to institutionalize slavery. Multiracial women in the U.S., who by their existence defy the construct of race, have the highest lifetime IPV exposure at 63.8%, followed by 57.7% of American Indian or Alaska Native women and 53.6% of Black women—these estimates include physical violence, contact sexual violence, and stalking by an intimate partner [8].

### 1.3. Traumatic Brain Injury (TBI) and Hypoxic–Anoxic Brain Injury (HAI)

Strangulation, and blows to the head, neck, and face are the most common physical violence reported by survivors of IPV [9,10,11]. These injuries place IPV survivors at higher risk for both traumatic brain injury (TBI) from rapid head acceleration/deceleration, focal blows, shearing forces, and or penetration and hypoxic–anoxic brain injury (HAI) from oxygen deprivation [12,13]. In their lifetime, 7.8% of U.S. women reported IPV-related impacts of broken bones or teeth, 7.0% of back or neck injuries, and 7.5% of head injuries [8]. A total of 7.2% of women reported being knocked out after getting hit, slammed against something, or choked—a conservative estimate of brain injury from an intimate partner. Although survivors are at risk for brain injuries from IPV assaults involving head trauma and oxygen deprivation, they are poorly represented in mainstream brain injury research [12,13].

Historically, brain injury research has focused on accidental TBIs from falls, motor vehicle crashes, and sports [12]. In fact, until 2023, TBI was considered an “accidental” injury caused by a sudden external force damaging the brain. The American Congress of Rehabilitation Medicine (ACRM) updated the diagnostic criteria for mild TBI (mTBI), in March of that year, to ensure the criteria applied across all sports, civilian trauma, and military settings [14]. To acknowledge TBIs caused by intentional assaults, including IPV, the word “accidental” was eliminated from references to injury events. Additionally, the new criteria focused the diagnostic criteria on mTBIs, recognizing that 90% of all TBIs are mild in severity. A person can be diagnosed with an mTBI if they have a plausible mechanism of energy transfer to the brain from an external force (Criterion 1) followed by *at least one of Criteria 2–5* and not being better accounted for by confounding factors (Criterion 6):

Criterion 2—at least one clinical sign (i.e., loss of consciousness, change in mental status, complete or partial amnesia, or other observed neurologic sign(s)); 

Criterion 3—two or more new acute symptoms from one or more categories (i.e., acute subjective alteration in mental status, physical symptoms, cognitive symptoms, or emotional symptoms);

Criterion 4—at least one clinical examination or laboratory finding (i.e., cognitive, balance or ocular motor impairment on acute clinical exam or elevated blood biomarkers indicative of intracranial injury); 

Criterion 5—neuroimaging indicative of trauma-related intracranial abnormalities.

Though neuroimaging is not necessary for diagnosis, if there is neuroimaging evidence the qualifier of “with neuroimaging evidence of structural intracranial injury” may be added to the diagnosis of mTBIs. The “mild” qualifier is removed from a TBI if there was a loss of consciousness greater than 30 min, if after 30 min the Glascow Coma Scale Score was less than 13, or post-traumatic amnesia occurred for more than 24 h.

A TBI results in heterogeneous injuries with both focal and diffuse components. Mass lesions, including focal contusion, and subdural or epidural hematomas may occur at the point of impact or elsewhere in the brain, such as opposite the impact site in coup–countercoup brain injuries [15]. Axonal or microvascular diffuse injuries affect wider regions of brain tissue [12]. For example, TBI can occur when the brain shifts and rotates inside the skull, resulting in lateral or rotational movement and the tearing of tissues and blood vessels. This type of damage can be seen in IPV but is often seen in child abuse cases as a result of vigorous shaking, resulting in abnormalities in axonal functioning [16]. A diffuse brain injury can be more difficult to detect than mass lesions using neuroimaging techniques [17].

Separately, HAI is studied most commonly in survivors of stroke or cardiac arrest, focusing on altered neurologic function [13,18]. Though there are no diagnostic criteria for HAI, it is defined as damage caused by the lack of oxygen to neural tissues [19]. Anoxic injury occurs when no oxygen reaches the brain tissue and can result after several minutes of hypoxia or reduced oxygen flow to tissues. When the oxygen supply to the brain is suboptimal, the brain enters a state of inadequate energy and cellular ion imbalance ensues [20]. In cases of recovery from cardiac arrest and stroke, a normal bioenergetic state will be restored. HAI is also characterized by a period of secondary cell death even after oxygen restoration, leading to delayed damage in the period following the oxygen-limiting event. With a few exceptions, unique mechanisms of damage to the brain caused by TBI and HAI have historically been studied independently. In the reality of the IPV context, they are often experienced together.

### 1.4. The Early Studies of Head Trauma and Strangulation Exposure from IPV

Studies in the late 1990s shed light on the issue of the pervasive exposure to head trauma among service-seeking DV survivors. Through a descriptive study of case reports of 35 women who entered DV shelters consecutively from November 1994 to January 1995, Monahan and O’Leary found 35% of battered women had experienced a head injury by an intimate partner [21]. Though they mention the exposure to head injury, a measure of the loss or alteration in consciousness was not obtained, as is the case in most research produced in tandem with colleagues working in clinical capacities.

By the turn of the decade, strangulation was also being introduced to peer-reviewed literature and clinical practice. Taliaferro, Mills, and Walker introduced five articles that appeared in a 2001 volume of *The Journal of Emergency Medicine* on the topic of strangulation and DV [22,23,24,25,26,27]. Though Taliaferro et al. [22] mentioned anoxic injury in the introduction, the articles primarily focused on strangulation and evidence collection as three of the articles were based on a review of 300 attempted strangulation cases submitted for misdemeanor prosecution to the San Diego City Attorney’s Office [23,24,25]. Injuries found in the non-fatal strangulation cases were similar to those in fatal DV assaults, however the limited training of police offices and prosecutors may have caused them to overlook corroborating symptom evidence of a survivor’s accusation of strangulation [23,25]. The first suggested protocol for the evaluation and treatment of survivors of non-fatal strangulation by DV is offered by [24]. In the same volume, Wilbur was one of the first researchers to document the pervasive exposure to strangulation experienced by DV survivors [26]. In this study, 62 patients were recruited from a hospital emergency department for being at risk for DV and referred to a local non-profit violence prevention center for survey. A total of 68% of the women surveyed were strangled at least once by an intimate partner (husband—55%, boyfriend—31%, and fiancé—5%), or by a mother, stranger, or friend (one each). However, only 29% of the those who were strangled sought medical care [26]. Finally, Smith and colleagues collected data in 2000 from 101 female subjects on a “study regarding violence” through both emergency rooms and DV shelters in multiple locations around the U.S. Participants were grouped by the number of strangulation attacks they had experienced at the hands of an abusive partner [27]. Findings revealed that individuals that experienced multiple strangulation attacks by the same abusive partner on separate occasions reported neck and throat injuries, psychological disorders, and neurologic disorders with more frequencies. Despite mentioning the increased frequency of neurologic disorders (i.e., dizziness, lightheadedness, headache, memory loss, vision changes, tinnitus, eyelid drop, weakness, facial droop, paralysis, loss of sensation, and muscle spasms) and including a definition for strangulation that mentioned a loss or restriction of blood flow to the brain, there was little discussion of HAI, specifically, being a long-term consequence of strangulation.

### 1.5. The Early Studies Naming Traumatic and Hypoxic Partner-Related Injury

In a sample of 53 women attending support groups at DV women’s shelters and community outreach programs in January of 1997, Jackson and colleagues found 82% of women had exposures to both severe shaking and having been hit in the head or face by an intimate partner [28]. Whereas 40% of the women in the sample reported ever having lost consciousness, meeting a diagnostic criterion for brain injury; 77% met criteria of post-concussive syndrome. A substantial number of survivors reported past-year and current struggles with distraction, concentration, memory. They also endorsed forgetting appointments and difficulty finding the right words.. They also shared ongoing issues with headaches, dizziness, trouble following directions, losing things, work becoming harder, and troubles in distracting environments.

In an early-2000s study conducted by Corrigan and colleagues of a multiracial sample of 46 women referred by the sexual assault–domestic violence (SA-DV) health staff at three local emergency departments, 30% respondents reported episodes of assault with a loss of consciousness, meeting a diagnostic criterion for brain injury, with another 10% unsure if they had lost consciousness [29]. Among all women, 67% reported new onset residual neurobehavioral sequela following IPV exposure consistent with the effects of a mTBI, including headache, dizziness, memory loss, and struggles with concentration and work or school performance. There were no differences in neurobehavioral symptoms reported between those reporting and not reporting a loss of consciousness. The SA-DV staff identified 35% of the respondents as potentially having sustained a mild brain injury, so they were referred to follow-up outpatient rehabilitation services. Based on these findings, authors recommended using a simple screening to identify possible mTBIs among survivors in early stages after the injury and offering treatment strategies (e.g., addressing physical complaints like headaches, vertigo, and sleep disturbances, along with support, cognitive restructuring, and antidepressant medication) in order to prevent the onset of disease and vocational problems from the worsened psychogenic components of post-concussive disorder when a brain injury is not properly diagnosed and treated.

Valera and Berenbaum conducted a unique study in the early 2000s in that they reported results on the more inclusive term “brain injury” and sought to capture information on mild, moderate, and severe TBI as well as “choke-inducted anoxia or hypoxia”. This is the first study that we are aware of where both types of injury common among IPV survivors are reported in the same study of the same sample. Ninety-nine women who had ever sustained any type of physical abuse by an intimate partner were recruited from DV shelters and programs to assist with protection orders, support groups for women with substance use histories and for couples to promote healthier communication, as well as friend referrals. In this purposefully selected community sample, 74% sustained a partner-related TBI at least once and 51% multiple times—with similar rates between shelter and non-sheltered survivors [30]. In addition, while 27% of battered women sustained partner-related “choke-inducted anoxia or hypoxia” once, 12% did so multiple times. Strangulation-induced brain injury was more common among the sheltered sample. By removing participants due to factors that may mask or confound the effects of partner-related TBI, a subset of 57 women were retained for a further analysis regarding the brain injury severity and associated symptoms. A brain injury was classified as mild if the loss of consciousness following an incident was less than 30 min and the post-traumatic amnesia was less than 24 h. A more severe injury was associated with the post-traumatic stress disorder (PTSD) symptomology as well as general distress, anhedonia depression, worry, and anxious arousal. Whereas a more severe brain injury was negatively associated with memory, learning, and cognitive flexibility.

### 1.6. Trauma, Strangulation, and Head Injury in Research and Practice

Despite Valera’s early use of the broader term “brain injury” to describe mild, moderate, and severe TBIs as well as “choke-induced anoxia or hypoxia”, much of the research in this area—particularly at the time of our investigation in 2019—remained siloed. At that time and in the years that followed, although this body of work has been invaluable for improving the safety and wellbeing of survivors, most published scholarship focused either on strangulation as a lethality risk factor or on the documentation of strangulation for prosecutorial purposes [24,25,26,30,31,32,33,34,35,36,37,38,39,40]. When brain injury was mentioned in relation to strangulation, though hypoxia or asphyxia may be called out, the resulting damage was often classified as TBI, not HAI [31,41,42,43,44,45,46,47,48,49,50,51,52,53,54,55]. At time of the data collection for the current study, Valera’s early study remained one of the few to bridge these areas, while the majority of the IPV-focused literature continued to address either IPV-related TBIs or non-fatal strangulation without explicitly identifying hypoxia as a mechanism of brain injury [31,41,42,43,44,45,46,47,48,49,50,51,52,53,54,55].

Importantly, though all the research has been valuable (including naming IPV as a source of BI and to demonstrate the impact and lethality risk of non-fatal strangulation in IPV), HAI was rarely named as a consequence of non-fatal strangulation, and few IPV-specific studies examined both TBI and HAI together. Research that does assess the prevalence of both mechanisms within the same study is only now re-emerging in the IPV, strangulation, and brain injury literature—nearly two decades later.

Until the last few years, when HAI was mentioned it was typically as a secondary injury following a TBI from IPV, or strangulation was reported to be a cause of TBI—but HAI as a unique kind of BI from strangulation and other forms of oxygen deprivation exposure by an intimate partner was not being discussed as a unique cause of IPV-related BIs [41,42,43]. There is a similar lack of connection in the strangulation literature, which is almost exclusively studied in the context of IPV. Psychological trauma, PTSD, and lethality threat are often discussed, yet brain injuries resulting from non-fatal strangulation are nearly absent [31,44]. Historically, research found that strangulation was specifically associated with a lack of consciousness, bladder control, and brain death [56]. The current literature indicates non-fatal strangulation events, such as the obstruction of the larynx or the occlusion of jugular veins and internal carotid arteries, may result in the triggering of the carotid sinus reflex and damage to the thyroid gland [57]. Bichard et al. [57] connects such strangulation-related events with indicators of brain damage, including a loss of consciousness, disordered speech and motor function, behavioral changes, and memory problems. Previous research has found that women who have experienced TBIs and strangulation report symptoms of alterations in neuropsychological functioning, alterations in consciousness, and post-concussive symptoms. However, there are few articles in the literature dissecting these separate types of injury when they co-occur and their complex interactions specifically in the context of IPV [58,59]. In addition, we recognize that survivors of IPV are more likely to experience other forms of family and interpersonal violence across the life course that may also target the head, neck, and airways and may result in brain injury from violence (B-VI). IPV survivors require an intentional focus on their unique circumstances when considering brain injury.

### 1.7. Context and Aims of Current Research

While research has advanced the documentation of IPV-related TBIs and non-fatal strangulation as separate concerns, the fact that the actual hypoxic injury resulting from strangulation may be a mediator that created an additional vulnerability for a heightened risk for these negative outcomes remains underexplored [51]. This disconnect is particularly concerning given that many IPV survivors report a repeated exposure to both TBIs and HAIs—often within the same violent episode and across multiple incidents over the life course. Furthermore, much of the foundational brain injury literature has been built on models derived from male populations, failing to capture the complex interplay between neurological injury, psychological trauma, and coercive control in survivors’ lives. In addition, much of the literature on brain injury, historically from animal modeling to community practice, has focused on TBIs (note: there are no diagnostic criteria for HAI) among men without modeling or understanding how a brain injury sustained in the context of ongoing psychological trauma and coercive control by an intimate partner could impact screening, access to services, rehabilitation, access to the social determinants of health, and other health-related outcomes. Despite this, in the context of IPV, the survivor’s brain has likely been exposed to both intentional TBIs and HAIs over the course of a lifetime and even within the context of the same incident [30,60]. The lack of representation of IPV survivors’ unique experience and context within brain injury research is one way in which health and research disparity gaps are perpetuated, detrimentally impacting IPV survivors overrepresented by populations subjected to oppression and intergenerational trauma.

This study is part of a larger community demonstration project funded by the United States (U.S.) federal government to build the capacity of domestic violence (DV) organizations to meet the needs of survivors presenting to services with mental health and TBI concerns. DV organizations in the U.S. provide services to survivors of the spectrum of family violence; however, most participants who access services present with IPV concerns. During the needs assessment portion of this project, our team discovered that most service-seeking survivors had exposures from intimate partners and others throughout the course of their life that could result in both intentionally inflicted HAIs and TBIs. In clarifying questions that survivors asked through the needs assessment interview, it became clear that survivors often experienced hypoxic and traumatic neurologic assaults in the same violent episode, and in other abuse settings over the course of their lifetime, yet our survey tools were not designed to capture information about the co-occurrence of HAIs and TBIs or other brain injuries sustained in other abuse settings [60]. The present study was implemented to try and fill both research and practice gaps and to better understand the lived experience of service-seeking DV survivors and the context in which partner-inflicted brain injuries (PIBIs) are sustained. Among service-seeking survivors we aimed to 1. explore the co-occurrence of strangulation and head trauma exposures from all abuse sources across the lifetime (though not necessarily within the same event), and within the first and last IPV incidents that could result in brain injury; and 2. establish the lifetime prevalence of probable partner-inflicted brain injury (PIBI), a term used to describe brain injuries from both traumatic and hypoxic neurologic insults, often experienced together and repeatedly, in the context of an abusive intimate relationship held together through psychological trauma and coercive control.

## 2. Materials and Methods

### 2.1. Survivor Interviews

Data were collected in 2019 from IPV survivors as part of the outcome evaluation of CARE, an advocacy intervention that raised the capacity of domestic violence service organizations (DVSOs) to provide trauma-informed care by addressing brain injuries [61]. Survivors 18 years of age or older participated in semi-structured interviews. Survivors were recruited from the 5 DVSOs that partnered in the evaluation of CARE—a federally funded project overseen by the Ohio Domestic Violence Network (ODVN) in partnership with The Ohio State University (OSU). The DVSOs provided a variety of services including clinical-, shelter-, and community-based support. Both rural and urban DVSOs participated. All study procedures were approved by the OSU Institutional Review Board.

### 2.2. Participants

The ODVN and the collaborating agencies identified two days on which OSU-trained research staff could come to the agency to conduct semi-structured interviews with survivors seeking services. OSU staff invited survivors to participate in immediate interviews or scheduled an interview during the two-day window when they were on site. Interviews lasted between 60 and 75 min, and participants received a gift card worth twenty dollars (USD).

### 2.3. Instrumentation

The semi-structured interview contained quantitative questions about survivor experiences. Questions focused on experiences with strangulation, head trauma, and the co-occurrence of these exposures. See the results section for questions asked, and response options posed to participants. To understand the lifetime exposure to specific types of assaults, questions included the following: “How many times in your life have you ever been choked or strangled?” Survivor experiences with medical services, as well as brain injury diagnosis post-strangulation and head trauma, were investigated. Co-occurrence questions explored lifetime, first, and last exposures as well as the timing of and the number of co-occurrences. A modified version of Valera and Berenbaum’s Brain Injury Severity Assessment (BISA) was used to assess probable brain injury **[30]**. Participant responses were recorded by the interviewer into Qualtrics.

### 2.4. Analysis

Data were summarized with frequencies and percentages or means and standard deviations (SD) as appropriate. As missing data were minimal, no imputation was performed, and results are presented based on available data. All data cleaning and analysis was conducted with SAS 9.4 (SAS Institute, Cary NC, USA).

*Probable PIBI—Loss of Consciousness (LOC) Classification*: Participants were classified as having a probable PIBI in their lifetime using the LOC standard if they responded “once”, “a few times”, or “too many times to remember” to the question “After anything your partner has ever done to you, did you ever black out or lose consciousness?” from the BISA measures [30]. This question was preceded by the root question statement “Now I want you to think about times when your current or former partner hit your head or strangled you”. The BISA question series was asked immediately following the question series capturing information on the co-occurrence of being hit in the head and being choked or strangled during the first and last IPV episode during which an intimate partner targeted the head or airways.

*Probable PIBI—Alteration in Consciousness (AIC) Classification*: Participants were classified as having a probable PIBI in their lifetime using the AIC standard if after being read the root question statement they responded “once”, “a few times”, or “too many times to remember” to at least one of the questions from the BISA measures, indicating an altered mental status following the event. This includes LOC but also includes other symptoms of AICs, including feeling dazed, confused, disoriented, or dizzy; seeing stars or spots; or having memory loss of what happened [30].

## 3. Results

### 3.1. Sample Characteristics

Forty-seven IPV survivors completed interviews. The majority were female (97.9%), non-Hispanic (93.6%), and white (88.6%) with a mean age of 38 years (Table 1). Half of the survivors had at least some college education, 17.0% had no current health insurance, and 42.6% were on Medicaid, jointly funded federal and state healthcare for the poor.

### 3.2. Lifetime Exposure to Strangulation and Intentional Head Trauma—All Sources

Most service-seeking IPV survivors had been intentionally choked or strangled at least once in their lifetime (83%)—this includes by an intimate partner or other person (e.g., parent, sibling, other relative, or peer); however, among those, only 28.2% sought medical services after an event and 15.8% were diagnosed with a brain injury as a result of being choked or strangled (Table 2). A similar pattern is present for intentional head trauma across the life course, with 87.2% of survivors ever having been hit in the head or made to have their head hit another object. Notably, nearly half of survivors (48.9%) reported having been hit in the head “Too many times to remember”. Of those reporting a lifetime exposure to intentional head trauma, nearly half sought medical services at least once after experiencing a direct blow to the head and almost all remembered receiving a concussion screening or neurological exam. Among those with a lifetime exposure to intentional head trauma, 36.6% had ever been diagnosed with a concussion or brain injury resulting from the intentional head trauma (Table 2).

### 3.3. Lifetime Exposure to Both Strangulation and/or Intentional Head Trauma—All Sources

The lifetime exposure to intentionally inflicted strangulation and head trauma, this includes by an intimate partner or other person, is summarized in Figure 1. A small proportion of service-seeking IPV survivors never experienced either strangulation or a direct hit to the head (10.6%), while an overwhelming majority experienced both (80.9%). A single survivor reported exposure to only strangulation, and three (6.4%) were exposed to only intentional head trauma (Table 2).

### 3.4. The Co-Occurrence of Head Trauma and Strangulation by an Intimate Partner

When including the first and last episode of a partner-inflicted neurologic insult, nearly half (48.9%) of all survivors experienced the co-occurrence of both head trauma and strangulation in the same incident (Table 3). On average, the first exposure to either strangulation or intentional head trauma by an intimate partner occurred at 23 years of age and the last exposure at 34 years of age (Table 3). The first exposures often involved either only head trauma (43.9%) or both head trauma and strangulation (43.9%), as shown in Figure 2; two (4.9%) of the participants only experienced strangulation and three (7.3%) did not know the type of neurologic insult that occurred on their first exposure. The last exposure to a neurologic insult experienced by survivors most commonly involved both hit(s) to the head and strangulation (37.5%) followed by direct hit(s) to the head only (32.5%) and strangulation only (25%); two (5%) of the survivors did not know what type(s) of neurologic insults occurred during their last exposure. There is a marked increase in the exposure to strangulation specifically at the time of the last neurologic insult by an intimate partner in comparison to the first time.

### 3.5. Probable Partner-Inflicted Brain Injury

When being asked to think about past events where a current or former intimate partner hit them in the head or strangled them, nearly 90% of participants endorsed at least one BISA concussion indicator and were classified as having a probable PIBI presently or at some point in the past based on the AIC classification. Over three in four survivors reported feeling dazed, confused, or disoriented at least once (Table 4) [8]. This was followed by seeing stars or spots (75.0%), feeling dizzy (75.0%), experiencing memory loss (68.9%), and blacking out or losing consciousness (62.2%). Alarmingly, for many survivors these PIBI indicators occurred repeatedly—13.3% of survivors lost consciousness, 27.3% saw stars or spots, and 17.8% had memory loss about what happened “Too many times to remember”.

## 4. Discussion

### 4.1. Reflection on Results

Our findings indicate that the majority of service-seeking IPV survivors in our region are at risk of having sustained intentionally inflicted brain injuries—both TBI and HAI—from various forms of violence experienced throughout their life course. Importantly, the specific assaults leading to each type of injury may not have occurred during the same violent event. Nearly half of the same survivors will have been targeted for both traumatic and hypoxic injury within the same violent episode by at least one intimate partner. Furthermore, nearly all survivors in our sample experienced at least one partner-inflicted brain injury (PIBI), whether from traumatic forces or strangulation. These injuries, often sustained in coercive, dominating, and terrorizing contexts, were reported by survivors accessing a range of advocacy services, including emergency shelter, case management, counseling, and health and criminal justice advocacy [1,2,3]. Both types of injury, along with the violent exposures that caused them, likely shape a survivor’s presentation and influence the kinds of accommodations needed just to access lifesaving services. Even if survivors do not directly disclose these experiences, it is crucial to offer universal education on head trauma and strangulation, including the impacts of resulting brain injuries. Organizations more likely to serve populations at an elevated risk for IPV must proactively adopt accommodations and procedures to address PIBIs and other forms of brain injury from violence (BI-V), whether sustained acutely and/or across the life course. Until a further inquiry can take place, it is reasonable—and potentially lifesaving—to assume that survivors may have sustained both TBIs and HAIs from IPV.

Over 8 in 10 participants in our sample of IPV survivors seeking advocacy services in Ohio (U.S.) were survivors of both intentional traumatic and hypoxic neurologic insults over the course of their lifetime, from all sources of violence including intimate partners, other family members, and peers, though they were not necessarily experienced during the same violent event. In our Ohio-based sample of survivors presenting to DV services, over 80% reported a lifetime exposure to *both* traumatic and hypoxic BI-V, which includes PIBIs. Nearly 5 in 10 of the service-seeking IPV survivors in our sample endorsed a concurrent exposure to both mechanisms of injury—hypoxic and traumatic—within their first and/or last incident of intimate partner violence targeting the head or airway. Additionally, 9 in 10 service-seeking IPV survivors were estimated to have sustained a brain injury from an intimate partner based on an alteration in their consciousness standard, and over 6 in 10 reported losing consciousness or blacking out due to a partner’s actions—a LOC estimate of a probable.

Most studies estimating the prevalence of TBI and HAI among IPV survivors have relied on small samples from organizations set up to address the safety and justice concerns of survivors, including DV shelters and non-residential services that provide support programs, counseling, and legal-, health-, or community-based advocacy. This reliance on convenience samples poses challenges to generalizability [62]. A recent (2022) scoping review identified 18 studies assessing the prevalence of BI among IPV survivors. Reported estimates of TBIs ranged from 28% to 100% and estimates of strangulation-related inferred brain injuries ranged from 27% to 56%, depending on the sample, measurement tools, and case definitions for brain injury [62].

Adhikari and colleagues recently examined IPV-related brain injury (IPV-BI), including the co-occurrent lifetime exposure to strangulation-related alterations in consciousness (S-AIC) and TBIs from intimate partners, among women who have experienced IPV. Unlike our sample, this was a pooled sample of intentionally selected women who experienced IPV in Canada, Columbia, Spain, and the U.S. recruited from both the community and IPV service organizations [59]. Their study assessed alterations in consciousness (AICs) using BISA measures and categorized exposures as either strangulation-related or traumatic. Finally, they classified IPV-related BIs through the AIC type as S-AICs and/or TBI. Approximately 67% of women who experienced IPV across these four countries had at least one IPV-related BI. Within that subsample of women with IPV-related BIs, 37% had sustained at least one S-AIC and at least one TBI from an intimate partner over time, whereas 2% had experienced S-AICs exclusively and 61% sustained TBIs exclusively. Despite differences in methodology, our findings align with Adhikari’s in showing that women survivors of IPV accessing community services are presenting with a history of both HAI and TBI. Both studies underscore the need for practitioners and researchers to capture sources of intentional injury that can lead to BI-V, because these wider exposures need to be considered when directing connections, care, and resources.

To our knowledge, we believe our study provides the first estimate of the co-occurrence of hypoxic and traumatic exposures leading to PIBIs within the same violent event. This finding highlights the need for scientific models and clinical approaches that acknowledge the complexity of dual-injury mechanisms sustained through both traumatic and anoxic causes of brain injury, occurring simultaneously. Everything from animal models to protocols to address strangulation and TBI in practice also need to be overhauled to account for this reality.

Although IPV survivors are situated in the crosshairs of these two distinct types of brain injury, they have seldom been represented in mainstream brain injury research outside of IPV-specific research [12,21,28,45,56,58,63]. Most brain injury research worldwide and public support at federal levels in the U.S. have focused through the lens of TBI [12]. Notably, in our study strangulation as a tactic to terrorize was experienced by more survivors during the last event (63%) in comparison to the first time (49%) they experienced head trauma or strangulation from an intimate partner. While only 5% of service-seeking survivors reported strangulation as the sole mechanism of injury in their first violent encounter, this increased to 25% during the last incident. Strangulation appears to escalate as a tactic in physical assaults over the course of an abusiverelationship. However, this pattern warrants further investigation, as our study did not ask participants whether they were referring to the same partner when reporting their first and last episodes of head trauma and/or strangulation. Prior research by Glass and colleagues has shown that strangulation significantly increases the risk of future death from IPV. Consequently, both research and clinical practice must prioritize recognizing and addressing hypoxic injury alongside TBI to better reflect the realities faced by survivors.

Supporting this assertion, worse outcomes have been observed among individuals who experience HAI in populations more frequently studied in brain injury research. This includes studies where hypoxic injury was compared with TBI only, or when both TBI and HAI were studied within the same individuals. Given the elevated risk of the severe functional impairment and death associated with strangulation—particularly when compared to TBI alone—it is imperative that both the study and clinical care of IPV-related brain injuries reflect the compounded and escalating risks over time posed by this terrorizing tactic [31,64,65,66,67].

Our findings support the co-occurrence of both traumatic and hypoxic injury mechanisms of injury among IPV survivors—both over the lifetime and within the same incident. Prior studies should be replicated using new animal models that account for repetitive and concurrent experiences of such injuries. Clinical and community-based research must evaluate the impact of concurrent traumatic and hypoxic neurologic insults on injury, healing, disability, effective rehabilitation, service access, outcomes, and social and health disparities, especially for IPV survivors—one of the most vulnerable, isolated, and invisible populations most likely to experience brain injury.

Our study is not without limitation. At the advisement of our community of practice advisory board, we limited the recall of specific IPV events to the first and last incident where the head or neck was targeted, or where strangulation was involved. Consequently, we did not ask about the worst incident or other incidents. We also focused solely on asking about strangulation as an exposure that can lead to HAI. Recent research in the United Kingdom has found that in a study of domestic abuse survivors seeking DV refuge, 75% have been held in a way in which they could not breathe—this included strangulation but also a partner’s use of water and waterboarding, choking, suffocation, or smothering with an object [68]. Both of these limitations have most likely led to an underreporting of the co-occurrence of head trauma and airway targeting in the same incident and of intentionally inflicted exposures that can lead to HAI in general. This also points to our need to implement validation and reliability studies to assure our current questions in the general brain injury space, and among researchers working in the area of brain injury from violence, are accurately representing abuse victims’ experiences of intentional violence that can result in TBIs and HAIs. These studies should also assure the current questions reflect accurate experiences across the life course and within episodes of violence, and are appropriate for use across diverse populations who may experience abuse differently.

Finally, with deep regret, we realized that we had mis-programed our survey in regard to collecting the person who inflicted intentional head trauma and strangulation in the section regarding the exposure across the lifetime and from all sources. Anecdotally, our interviewers recalled that strangulation was a tactic often used by intimate partners, whereas there was more variation in the source of injury in the head trauma category (i.e., intimate partners, other family members, peers, etc.). Though we cannot say this with confidence, as our data from this question are unreliable, this pattern of exposure would mirror the source exposure for intentional hypoxic and traumatic brain injuries (BI-Vs) that the first author found in a research study involving U.S. youth experiencing homelessness (YEH), a population from 14 to 24 years of age with unstable or no housing [69]. Up to three in four YEH have been targeted for IPV in their life, in addition to being at a heightened risk when compared to their housed peers for fleeing homes with violence, experiences with human trafficking, and peer violence [70,71,72]. In Nemeth’s study, data were collected on intentional and accidental sources of HAI and TBI, separately. Participants were classified as having a probable brain injury if they responded “yes” on any of the BISA indicators following each type of injury, so using the AIC classification for BI. While 65% (61/96) of the sample reported at least one exposure to an oxygen deprivation event, 51% (49/96) reported an intentional event where they had been choked, strangled, or held in a way in which they could not breathe. All those with an intentional hypoxia exposure met the classification for having a probable brain injury. Among those with intentional HAIs, 63% reported ever being targeted by an intimate partner, where 30% reported exposure from a peer, and 28% both from a parent/guardian or from a brother/sister—notably, some participants were held in a way they could not breathe from multiple sources. In the same sample, 87% (83/96) reported at least one traumatic exposure that could lead to TBI, and 73% (69/96) reported an intentional head trauma event, including both being hit and shaken violently.

Despite these limitations, this is the first study that we are aware of that has not only considered the co-occurrence of hypoxic and traumatic injuries across time but also considered the co-occurrence of hypoxic and traumatic injuries from all sources of intentional injury and during the same event of intimate partner violence.

### 4.2. Partner-Inflicted Brain Injury

IPV survivors are experiencing both head trauma and strangulation, which cause damage via differing injury mechanisms to the same neural systems, over time and in the same violent event [30,59,60,69]. And though there have been some efforts to include concurrent TBI, strangulation, and hypoxic injury in research and practice settings, we need a unifying language to prompt clinicians, researchers, and the general community to consider psychological trauma, TBI, strangulation, and HAI every time they are working with or studying a survivor of IPV—including in settings and among populations where IPV is more likely to impact patients, clients, and research subjects [30,59,60,61,69,73,74]. Therefore, a comprehensive term that encompasses both head trauma and oxygen-restricting events, including strangulation, in the unique context of IPV, is needed. Using scientifically correct terminology is necessary to clarify the presence of two distinct injury mechanisms, encourage research in tandem with one another, and realize implications for identifying and treating such injuries in survivors of IPV.

We propose the term partner-inflicted brain injury (PIBI) to describe this widespread, often undetected, and severely understudied public health crisis [48]. A PIBI is defined as the physiological disruption of normal brain function caused by intentional, often concurrent and repeated, traumatic and hypoxic neurologic insults by an intimate partner. PIBI occurs within the context of an abusive relationship held together through psychological trauma, coercive control, and threats of escalating physical violence intended to terrorize and subdue. The term incorporates TBI and HAI, reflects intentionality, emphasizes the psychological impact of the perpetrator being an intimate partner, and recognizes that violent episodes causing brain injury occur within an ongoing abusive context where psychological trauma is ever present.

PIBI encompasses TBI, which is already being studied in survivors and incorporates strangulation and resulting HAIs. Due to the frequent and often escalating violence related to the coercive control seen in IPV, survivors are at an increasing risk of multiple, concurrent, and repetitive injuries via both mechanisms over time [49,51]. Populations exposed to repeated head traumas that have been the subject of more studies show cumulative damage and negative outcomes [30,50,52]. In addition, oxygen deprivation in brain tissue has been shown to cause rapid cell death, and the existing TBI research suggests that brain hypoxia and anoxia is associated with worse outcomes and stunted recovery [64,65,66,67]. Understanding TBI, HAI, and their concurrence or combination over time are all vitally important to our understanding of this phenomenon and the provision of appropriate treatments.

In addition to the two separate mechanisms, their interaction, and the repetitive nature of IPV instances targeting the head, neck, and airway, there are several other reasons that PIBI is unique and must be accurately defined. Bystanders who can identify an injury and help immediately are largely absent in IPV situations. Survivors are often unable to seek medical care after sustaining a head injury and must attempt to recover in an environment of ongoing coercive control, violence, and stress [53,54]. In the IPV setting, survivors are frequently unable to adhere to suggested treatments for an injured brain, such as rest and a reduction in stimuli [75]. Furthermore, unlike in sports injury, there is no “return to play” instruction—a graduated stepwise rehabilitation strategy [76]. For the perspective of a survivor of IPV, this might be thought of as a survivor-centered and professionally supported rehabilitation strategy to “progress to safety, stability, healing, and wellness”.

In IPV, brain injuries are the result of violence perpetrated on purpose by someone intimate to the survivor. This causes psychological trauma distinct from the physical trauma of sustaining a brain injury from an accidental source, such as through sport participation or from a workplace hazard. The traumatically induced stress, generated by the intentional and interpersonal nature of interactions in the context of IPV, is unique and merits its own investigation. At the same time, we recognize that the PIBI belongs under a larger umbrella of brain injuries from power-based violence (BI-V), which would also include peer violence, family violence, child or elder abuse, and institutionalized violence—all of which may compound the psychological and physical impacts of PIBI [63,77,78].

IPV-related injuries and their presentations may be masked by or attributed to psychological trauma or the IPV experience itself—a well-accepted, alternative explanation absent in other populations experiencing brain injuries. The inclusion of PIBI to the current understanding of IPV survivorship and trauma-informed care is critical for improved health and psychological outcomes. PIBI is often underdiagnosed because survivors and professionals are not making the connection between IPV and brain injury [26]. The results of a 2019 study revealed that, although those in the sample at risk for TBI sought help from a professional, providers did not discuss TBI with the survivor during office visits [55]. Survivors may not be aware of the signs or symptoms of TBI or HAI, and many professionals are not considering brain injuries and associated symptoms or behaviors in relation to IPV when working with survivors [47,60,79]. Presently, although psychological trauma is recognized by advocates and healthcare workers as a result of IPV, neurologic trauma is not. IPV survivors suffering from mood disorders, PTSD, substance use, and suicidal ideation, for example, may not be able to fully access services or experience symptom resolution due to their unidentified neurologic injury [30].

Psychological trauma and brain injury can both change the way people feel, think, and act and result in executive function dysregulation, emotion dysregulation, memory struggles, and sleep disturbances [12,13,14,30,45,46,49,79]. Historically, within organizations addressing the needs of survivors of IPV, these manifestations have solely been attributed to the neurobiology of psychological trauma. However, by not attending to neurologic trauma, the progression to healing may be slowed or prevented entirely. Brain injuries also cause chronic pain, headaches, hearing and vision problems, balance issues, and language processing difficulties [12,13,14]. Critical in IPV settings, brain injuries are also known to impact a person’s ability to appraise threats, plan, and follow through—all critical to lifesaving actions by survivors in acute IPV settings where they are often separated from social supports. In addition, when service providers misdiagnose or underdiagnose brain injury, they miss opportunities to provide accommodations for brain injured survivors. This can lead to service failure; increasing a survivor’s risk of additional assaults; a lost opportunity for rehabilitation; and poorer outcomes. This reality highlights the necessity for IPV services to consider PIBI—which demands advocates incorporate an understanding of the impacts of both neurologic and psychological trauma experiences on survivors’ current presentation, especially when planning a way forward to survivor-defined goals, including those for safety, action, and healing.

### 4.3. CARE (Connect, Acknowledge, Respond, Evaluate): An Intervention to Address PIBI

Though intentionally inflicted brain injuries may result from either acute or distal exposures, individuals who experience both hypoxic and traumatic brain injuries face greater challenges in recovery and long-term functioning [64,65,66]. In the acute setting, clinicians may need to refer survivors to health services immediately to prevent additional harm [80]. However, even a mild brain injury may have lifelong impacts and need to be managed as a chronic health condition [66]. Though trauma-informed service delivery models have been adopted to guide practices in most advocacy settings working with IPV survivors, the neurologic injury from head trauma, strangulation, and other forms of oxygen deprivation was not historically addressed in trauma-informed care [61]. Service-seeking IPV survivors living with brain injuries will most likely need accommodations and support to access and participate in presenting services, including advocacy, health, criminal justice, or social services, and in moving forward with both survivor-defined and service system goals. Despite this, specific screening tools to account for concurrent and repeated traumatic and hypoxic brain injury through intentional mechanisms still need to be refined and validated among IPV survivors, and interventions need to be refined and validated among IPV survivors, and interventions need to be developed and modified for coordinated use across service systems to achieve this vision.

Through a partnership between The Ohio State University and the Ohio Domestic Violence Network, we have created the CARE program (Connect, Acknowledge, Respond, Evaluate), the first evidence-based, brain injury-informed organizational enhancement to trauma-informed IPV advocacy service delivery that focuses on universal education and accommodation [61,73,74,81]. The purpose of CARE is for organizations to purposefully address the medical, psychological, safety, and service access needs of IPV survivors, in general, by attending to PIBIs specifically. CARE recognizes the following: (1) PIBI sustained in the context of ongoing trauma can impact survivors’ thoughts, feelings, health, and ability to function, both after an acute event and across the life course; (2) PIBI is central to many survivors’ experiences and may contribute to challenges that must be accommodated within services in order for organizations to fulfill their purpose in supporting survivors; and (3) survivors deserve to have the consequences of brain injuries from violence considered in the very complex web of circumstances they must navigate, no matter where they are in the process of engaging and moving forward with services to address their presenting concerns. The impact of PIBI, or other exposures to BI-V, needs to be addressed and accommodated as survivors access safety, health, civil or criminal justice, social services, or the social determinants of health.

The CARE program provides practitioners with a process (The CARE Framework) and resources (The CARE Tools) to flexibly implement brain injury-enhanced trauma-informed care, provide education, implement accommodations, and make referrals when appropriate to support survivors in meeting their service goals [61]. The CARE Framework is implemented through a cyclical process and asks professionals to achieve the following: Connect genuinely with survivors;Acknowledge that head trauma and strangulation are common forms of IPV assault and can have physical, cognitive, emotional, and behavioral consequences immediately, but also years later;Respond by accommodating needs related to brain injury, strangulation, and mental health challenges in services and provide effective, accessible referrals and advocacy for individuals who need additional care;Evaluate if accommodations and referrals were effective, reassess needs, and respond accordingly.

The use of the CARE Tools has been an integral part of the way in which domestic violence service organization providers have implemented the CARE Framework in practice [48,50]. A process evaluation has revealed that advocates assisting IPV survivors are using the CARE Tools:To connect with survivors on the topics of head trauma, strangulation, and mental health concerns (e.g., suicide ideation, emotion lability, and substance use), to facilitate survivors’ self-advocacy and connection to other service systems (e.g., medical and courts), and for self-connection and self-care to address historical and secondary trauma that service providers have experienced;To acknowledge and normalize that head trauma, strangulation, and mental health struggles are common experiences among domestic violence survivors and recognize that ongoing education and self-care are necessary for advocates;To provide a more holistic approach to trauma-informed care through accommodations within services and referrals to other safety, health, justice, and social service systems so survivors can better access and benefit from services, and self-advocate, often by directly using the CARE Tools.

CARE is an evidence-based program that has raised the capacity of organizations to deliver trauma-informed care, and to address mental health, suicide, substance use, head trauma, and strangulation, because staff felt empowered to address brain injuries, re-frame survivor behavior as “can’t” not “won’t”, and offer accommodations and functional support [61,74]. Professionals working with IPV survivors have shared that their organization’s implementation of CARE in service delivery has allowed them to perform their jobs better—and this is paramount to retaining employees in a service system with higher burn-out rates and turnover [74]. Though initially developed through a partnership with local IPV service agencies in Ohio (U.S.), other providers working with survivors in a variety of settings and who are committed to implementing trauma-informed practices that center the needs of each survivor might consider adopting CARE for their use [61]. All CARE Tools are available free for download at the Center on Partner-Inflicted Brain Injury, a program of the Ohio Domestic Violence Network, at https://www.odvn.org/brain-injury/ URL (accessed on 5 May 2023). This includes educational tools, including a head injury education card in English and Spanish, an overview and booklet on invisible head injuries from head trauma and strangulation, and the Just Breathe guide to track plans and recovery. It also includes tools for service providers to implement CARE in practice, including PIBI organizational promising practices, CHATS, accommodation suggestions, and how to make groups effective for clients with cognitive impairments.

### 4.4. The Brain Injury from Violence CARE Alliance: Adapting CARE with the Community

Through continued work and expanded university–community partnerships, including the development of the Brain Injury from Violence CARE Alliance (BI-V CARE Alliance), we are in the formative stages of adapting and optimizing the CARE intervention for use in the justice system, community health centers that provide wholistic health services to the poor in the U.S., substance use treatment agencies, and organizations serving youth and young adults experiencing homelessness. In addition, we are also engaging organizations whose primary focus is to work with populations at risk of other forms of BI-V, including survivors of sexual violence and human trafficking—with hopes that survivors of all forms of power-based violence may receive the appropriate identification of and accommodations for potential TBIs and HAIs at places they go to receive life-promoting care, resources, support, and rehabilitation. Finally, we are also working to better understand how to modify CARE for use with justice-involved individuals, with the acknowledgement that although most survivors of violence do not go on to use violence, those who do use violence are more likely to have experienced abuse themselves. We recognize to prevent violence we must address BI-V and provide education and accommodations for those who have sustained BI-V throughout the life course. The BI-V CARE Alliance was founded in 2023 to bring together survivors, families, advocates, and professionals to understand and address brain injuries from violence. Our vision is to bring systematic change to address brain injuries from violence and ensure survivors and all affected have accessible resources to heal, feel empowered, and thrive. For more information about the BI-V CARE Alliance, and to receive communications, please email CAREAlliance@osu.edu.

## 5. Conclusions

This study demonstrated that service-seeking survivors of IPV experience repetitive and/or concurrent exposures to intentional strangulation and head trauma through the life course and by intimate partners within the same episode of physical violence. When alterations in neurologic function occur following such mechanisms of exposure, hypoxic/anoxic (HAI) and traumatic (TBI) brain injuries result, respectively. Because these two types of brain injury have not been extensively studied as co-occurrences, let alone in the context of the larger continuum of interpersonal trauma across the life course, it is unclear how much of the existing TBI research and the clinical practice paradigm can be applied to the experiences of IPV survivors [59]. By not equally attending to psychological trauma, TBI, HAI, and other injuries that can result from non-fatal strangulation, researchers and service providers across all settings are prevented from exploring the manifestations of psychological and neurologic trauma that impact the way IPV survivors think, feel, and act. Ignorance to the possibility of either type of trauma in any organization where IPV survivors present could both impede survivors from moving towards their goals and result in organizations and institutions not being able to achieve their purpose.

We suggest the term partner-inflicted brain injury (PIBI) to prompt researchers and clinicians to consider the physiological disruption of normal brain function caused by hypoxic and traumatic neurologic insults—inflicted intentionally, repeatedly, and often concurrently by intimate partners—and to consider that these brain injuries occur within the context of IPV where coercive control is used to terrorize and isolate survivors and among populations more likely to be the target of institutionalized power-based violence. We encourage the use of CARE (Connect, Acknowledge, Respond, Evaluate) in settings where IPV survivors participate in services [61,73,74,81]. CARE is a brain-injury aware enhancement to trauma-informed service delivery that focuses on PIBI universal education and accommodation. Organizations serving IPV survivors that have adopted CARE improved trauma-informed services, overall, including attending to strangulation, head trauma, suicide ideation, substance use, and other mental health concerns [61]. Improvements occurred because staff were able to recognize the impact of PIBI, reinterpret client behavior as “can’t” not “won’t”, and provide functional and social supports [73,74,81]. Anyone engaged in IPV knowledge production or service support must do more to illuminate PIBI—as a lifesaving act. The need for new terminology and characterization, interdisciplinary research, and the application of findings to practice is long overdue across brain injury and IPV settings.

## Figures and Tables

**Figure 1 brainsci-15-00524-f001:**
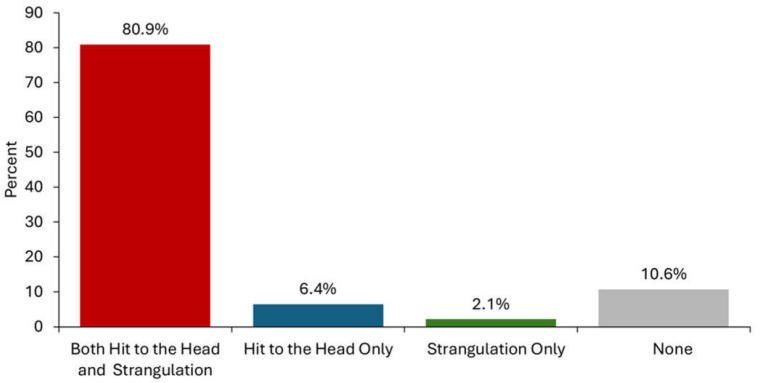
Lifetime exposure to intentionally inflicted head trauma and/or strangulation from all sources, including intimate partners and others (e.g., sibling and peer) (*n* = 47).

**Figure 2 brainsci-15-00524-f002:**
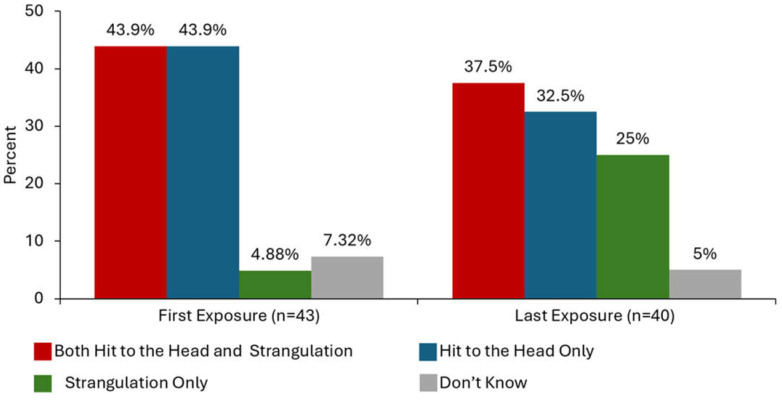
The co-occurrence of intentional head trauma and strangulation in the first and last incidence of physical IPV in which a partner targeted the head and airway.

**Table 1 brainsci-15-00524-t001:** Sample characteristics (study location: Ohio, U.S.).

Sample Characteristics (Study Location: Ohio, U.S.).		
	Service-Seeking IPV Survivors (*n* = 47)	
Age (mean, SD) [*n* = 43]	38.32	11.26
Highest grade level (*n*, %) [*n* = 46]		
Less than high school	6	13.04
High school diploma/GED	17	36.96
Some college	11	23.91
Associate degree or higher	12	26.09
Current health insurance coverage (*n*, %) [*n* = 47]		
None	8	17.02
Medicaid	20	42.55
Other	19	40.43

**Table 2 brainsci-15-00524-t002:** Lifetime strangulation and intentional head trauma—all sources: exposure, medical care, and brain injury diagnosis.

Lifetime Strangulation and Intentional Head Trauma—All Sources: Exposure, Medical Care, and Brain Injury Diagnosis.		
**2.A. Intentional Strangulation–All Sources (e.g., partner, sibling, and peer) ***		
How many times in your life have you ever been choked or strangled? (*n*, %) [*n* = 47]		
Never	8	17.02
Once	5	10.64
A few times	18	38.30
Too many times to remember	16	34.04
[Among those with lifetime strangulation exposure] have you ever sought medical services after being choked or strangled? (*n*, %) [*n* = 39]		
Never	28	71.79
Once	5	12.82
A few times	4	10.26
Too many times to remember	2	5.13
[Among those with lifetime strangulation exposure] how many times have you been diagnosed with a brain injury resulting from being choked or strangled? (*n*, %) [*n* = 38]		
Never	32	84.21
Once	4	10.53
A few times	2	5.26
Too many times to remember	0	0.00
*Table note.* (*) = Due to a skip pattern error, we are not presenting data on who has committed intentional strangulation, asked as the second question in series 2.A.		
**2.B. Intentionally Inflicted Head Trauma—All Sources ****		
How many times in your life have you ever been hit in the head or were made to have your head hit another object? (*n*, %) [*n* = 47]		
Never	6	12.77
Once	5	10.64
A few times	13	27.66
Too many times to remember	23	48.94
[Among those with lifetime exposure to intentional head trauma] have you ever sought medical services after you were hit in the head or were made to have your head hit another object? (*n*, %) [*n* = 41]		
Never	22	53.66
Once	9	21.95
A few times	6	14.63
Too many times to remember	4	9.76
[Among those who sought medical services] did you receive a concussion screening or neurological exam when you sought medical services? (*n*, %) [*n* = 18]		
No	1	5.56
Yes	17	94.44
[Among those with lifetime exposure to intentional head trauma] how many times have you been diagnosed with a concussion or brain injury resulting from being hit in the head or having your head hit another object? (*n*, %) [*n* = 41]		
Never	24	58.54
Once	7	17.07
A few times	6	14.63
Too many times to remember	2	4.88
Do not know	2	4.88
*Table note.* (**) = Due to a skip pattern error, we are not presenting data on who has committed intentional head trauma, originally asked as the second question in series 2.B.		
**2.C. Lifetime Exposures to Events That Can Cause Brain Injury from Violence (BI-V)** **—All Sources** (*n*, %) [*n* = 47]		
None	5	10.64
Strangulation Only	1	2.13
Intentional Head Trauma Only	3	6.38
Both Strangulation and Intentional Head Trauma	38	80.85

**Table 3 brainsci-15-00524-t003:** Partner-inflicted head trauma and strangulation—first and last exposure.

Partner-Inflicted Head Trauma and Strangulation—First and Last Exposure.		
**First Exposure to Partner-Inflicted Head Trauma and Strangulation** (*n*, %) [*n* = 41]		
Hit in the head only	10	24.39
Hit once in the head and choked or strangled at the same time	7	17.07
Hit multiple times in the head only	8	19.51
Hit multiple times in the head and choked or strangled at the same time	11	26.83
Choked or strangled only	2	4.88
Do not know	3	7.32
Age of First Exposure (mean, SD) [*n* = 38]		
	23.11	8.18
**Last Exposure to Partner-Inflicted Head Trauma and Strangulation** (*n*, %) [*n* = 40]		
Hit in the head only	7	17.50
Hit once in the head and choked or strangled at the same time	4	10.00
Hit multiple times in the head only	6	15.00
Hit multiple times in the head and choked or strangled at the same time	11	27.50
Choked or strangled only	10	25.00
Do not know	2	5.00
Age of Last Exposure (mean, SD) [*n* = 37]		
	34.73	10.39
**Co-Occurrence of Head Trauma and Strangulation by an Intimate Partner During First and/or Last Exposure When the Head or Airway was Targeted** (*n*, %) [*n* = 45]		
	22	48.89

**Table 4 brainsci-15-00524-t004:** Brain Injury Severity Assessment (BISA)—measuring loss of and alteration in consciousness to assess probable PIBI among service-seeking survivors*.

**Brain Injury Severity Assessment (BISA)—Measuring Loss of and Alteration in Consciousness to Assess Probable PIBI Among Service-Seeking Survivors ***		
**Now I Want You to Think About Times When Your Current or Former Partner Hit Your Head or Strangled You.**	Survivors	
	*n*	*%*
*After anything your partner has ever done to you, did you ever:* Black out or lose consciousness? [*n* = 45]		
Never	17	37.78
Once	8	17.78
A few times	14	31.11
Too many times to remember	6	13.33
Feel dazed or confused or disoriented? [*n* = 44]		
Never	10	22.73
Once	3	6.82
A few times	17	38.64
Too many times to remember	14	31.82
Have memory loss about what happened? [*n* = 45]		
Never	14	31.11
Once	5	11.11
A few times	18	40.00
Too many times to remember	8	17.78
See stars or spots? [*n* = 44]		
Never	11	25.00
Once	3	6.82
A few times	18	40.91
Too many times to remember	12	27.27
Feel dizzy? [*n* = 44]		
Never	11	25.00
Once	5	11.36
A few times	17	38.64
Too many times to remember	11	25.00
**BISA Indicator** [*n* = 45]		
No indicator	5	11.11
At least one indicator	40	88.89
**Probable PIBI** [*n* = 45]		
Loss of Consciousness (LOC) Criterion “Yes” to black out or lose consciousness	28	62.22
Alterations in Consciousness (AICs) Criterion “No” to black out or lose consciousness AND *At least two of the below*: Dazed, confused, or disoriented; Memory loss about what happened; See stars or spots; Feel dizzy	9	20.00
** ** **Probable PIBI based on LOC or AIC Criterion**	37	82.22
*Table note.* (*) = Asked of all survivors, regardless of whether they reported experiencing strangulation or direct hits to the head.		

## Data Availability

The original data and codebook presented in the study are openly available in GitHub at https://github.com/nemeth37/Partner-Inflicted_Brain_Injury. Uploaded by authors on 4 June 2024.

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
