# Peer review of "Partner-Inflicted Brain Injury: Intentional, Concurrent, and Repeated Traumatic and Hypoxic Neurologic Insults"

_brainsci, 2025, doi:10.3390/brainsci15050524_

Round 1
Reviewer 1 Report
Comments and Suggestions for Authors
The authors report interesting work in the field of IPV. The contribution results in a well-written and organized paper, with analysis adguate to the research aims/objectives. I only ask the authors for a minor revision:
- Report more sociodemographic data in the abstract and clearly state the nationality of the study as early as the abstract.
- Contextualize the topic of IPV in your cultural context by reporting some prevalence data.
- Increase the discussion of the practical implications of your study.
Reviewer 2 Report
Comments and Suggestions for Authors
The authors have submitted a manuscript describing partner-inflicted brain injury. The manuscript is well conceived and well written. The term, "partner-inflicted brain injury (PIBI) was offered to use for this population, including those who experienced a TBI and HAI (hypoxic-anoxic injury). Overall, it is well done. One question -although most of these injuries were done by partners who know the injured person, what do the authors have to say if this type of injury happened outside of the partner. For example, an injury caused by a friend, the environment or other ways I person can experience this type of violence/injury
Comments on the Quality of English LanguageA review of English grammar is suggested.
Reviewer 3 Report
Comments and Suggestions for Authors
General comments:
The paper addresses a complex and under-documented topic in scientific literature: Partner-Inflicted Brain Injury, cooccurrence of HAI and TBI is analyzed. The data come from semi-structured interviews about IPV survivor experiences. On the basis of only the interviews, a new type of Traumatic Brain Injuries (TBI), “partner-inflicted brain injury” (PIBI) has been proposed. The existence of PIBI, although possible and even probable, is not supported by the data (some additional material correlate would be required). The article has its merits as it explores an area with limited available literature, making it a potentially valuable contribution. However, it presents two significant flaws:
1. Lack of Evidence of TBI in IPV Survivors: There is no concrete material evidence provided that IPV survivors suffer or have suffered from any form of TBI, whether temporary or permanent.
2. Need for Physical Evidence to Corroborate PIBI: Although the proposition of the existence of PIBI is intriguing, additional physical evidence (such as medical tests or imaging) is required to confirm its existence and epidemiology among IPV survivors.
Specific comments:
- Line 38: Reference [9] mentions that injuries can occur “at the site of a direct application of blunt force,” but this is not the most common occurrence. Additionally, the review fails to mention Acute Subdural Hematoma as a consequence of blows, which can occur at a different location than the site of impact. This section should be improved to include these considerations.
- Line 75: It is mentioned that 70 million people suffer from some form of TBI each year for various reasons (García-Vilana et al., 2021). The low representation of IPV-related TBI is not surprising, as it is difficult to estimate compared to other causes of TBI.
-Line 115: The statistical analysis is limited to percentages and averages. It would be desirable to perform correlation analyses between different variables, as they are clearly not independent. Additionally, a Principal Component Analysis (PCA) could help identify case groups by reducing data dimensionality.
-Table 1: The table should consider a non-U.S. perspective. Outside the U.S., it is uncommon to use tables that include biological factors (gender), cultural factors (Hispanic vs. non-Hispanic), and phenotypic factors (White, Black). The concept of “race” as used by the United States Census Bureau lacks scientific utility outside U.S. contexts. I suggest clarifying the relevance of these factors or removing them.
-Section 3.4: This section is critical. The concussion symptoms described do not necessarily indicate a temporary or permanent injury associable with TBI. The brain depolarization causing blackout, loss of consciousness, or disorientation cannot be taken as effective evidence of TBI, although TBI can certainly cause such symptoms.
- Discussion: The presented data do not support the conclusion that PIBI exists without material correlates, medical tests, or imaging evidence. While it is plausible that PIBI exists and can be diagnosed, concrete material evidence is needed to assert its existence in this context.
Final comment:
The article reviews a potentially impactful area of study but requires improvement in several critical aspects. Additional material evidence is needed to support conclusions about PIBI, and a more robust statistical analysis is warranted. Moreover, it should consider an international perspective in its data presentation.
Reviewer 4 Report
Comments and Suggestions for Authors
Although the article presents interesting and, above all, alarming results as a consequence of violence in the couple, the implications it has in the field of psychology are not very clear. The article discloses the physical consequences of abuse, but does not offer data on the actions to be taken, which should be divided into two parts. An intervention with the victim from a medical and psychological point of view, and an intervention with the aggressor from a criminal point of view. It would be advisable for the authors to focus the results of their article in these sections.
Round 2
Reviewer 4 Report
Comments and Suggestions for Authors
The authors have carried out a complete revision of the study, improving it considerably. They have responded effectively to the requests made, so it is considered that the article can be published in current version. I congratulate the authors for the work and effort made.
Author Response
Thank you!